# User Experience of Virtual Human and Immersive Virtual Reality Role-Playing in Psychological Testing and Assessment: A Case Study of ‘EmpathyVR’

**DOI:** 10.3390/s25092719

**Published:** 2025-04-25

**Authors:** Sunny Thapa Magar, Haejung Suk, Teemu H. Laine

**Affiliations:** Department of Digital Media, Ajou University, Suwon 16499, Republic of Korea; sunnythapa@ajou.ac.kr (S.T.M.); teemu@ubilife.net (T.H.L.)

**Keywords:** user experience, virtual human, virtual reality, role-playing, psychological testing and assessment, empathy type

## Abstract

Recent immersive virtual reality (IVR) technologies provide users with an enhanced sense of spatial and social presence by integrating various modern technologies into virtual spaces and virtual humans (VHs). Researchers and practitioners in psychology are attempting to understand the psychological processes underlying human behavior by allowing users to engage in realistic experiences within illusions enabled by IVR technologies. This study examined the user experience of role-playing with VHs in the context of IVR-based psychological testing and assessment (PTA) with a focus on EmpathyVR, an IVR-based empathy-type assessment tool developed in an interdisciplinary project. This study aimed to evaluate the advantages and disadvantages of integrating IVR-based role-playing with VHs into PTA by examining user immersion, embodiment, and satisfaction. A mixed-method approach was used to collect data from 99 Korean adolescents. While the participants reported high levels of immersion and satisfaction, the sense of embodiment varied across the correspondents, suggesting that users may have had disparate experiences in terms of their connection to the virtual body. This study highlights the potential of IVR-based role-playing with VHs to enhance PTA, particularly in empathy-related assessments, while underscoring areas for improvement in user adaptation and VH realism. The results suggest that IVR experiences based on role-playing with VHs may be feasible for PTA to advance user experience and engagement.

## 1. Introduction

The digital age is driving significant shifts in the field of psychological therapy [1,2], with psychological testing and assessment (PTA) playing an essential part in these transformations. One of the most significant changes introduced by technology is the use of immersive virtual reality (IVR) in PTA, which allows the user to experience places and situations that they otherwise may not have experienced cognitively or emotionally. Continuous use of IVR technology and tracking devices to collect and transmit vast amounts of behavioral data is anticipated to become a reality in the near future [3]. The interactivity and sensor-based behavioral tracking technology of IVR will have a major impact on PTA, which goes beyond the social phenomenon or industrial trend that technological breakthroughs inevitably increase people’s desire to reflect on themselves or others. Many studies have found advantages in investigating social distance and human behavior through social presence using immersive virtual environments [4,5,6].

Traditional PTA methods rely on self-reporting, which raises concerns about the validity of causal conclusions owing to several factors, including social desirability bias [7], memory limitations, and recall bias [8], lack of self-awareness [9], and cultural and linguistic differences [7,10]. In addition, traditional PTA methods include systematic response bias, method variance, single-method bias, and the psychometric properties (reliability and validity) of questionnaire scales [11]. Simms (2019) demonstrated the poor psychometric accuracy of response scales containing two to five point options [12]. Despite the benefits of technology and concerns regarding traditional methods of conducting PTA, research shows that integrating IVR into mainstream PTA is hindered by diverse challenges such as cost, accessibility, and user adaptation [13,14]. IVR role-playing-based scales, however, are advantageous in their ability to provide real-time experiences and collect instant responses regarding scenario events [15]. Therefore, the potential remains if the issues of cost, accessibility, and adaptability are addressed through industry solutions. The authors and their interdisciplinary research team demonstrated the efficacy of utilizing scales based on user role-playing experiences as the foundation for self-determination in IVR scenarios [15].

The strength of IVR lies not only in its ability to provide spatial illusions for tele-presence, but also in the user’s ability to interact with virtual humans (VHs). IVR role-playing utilizes VHs unless the virtual avatars are played by real humans. VHs are embodied agents with a human-like appearance [5,16] and provide a representation of humans through computer graphics dummies in a virtual space that can simulate human movements and evaluate workloads [17]. They mimic the human skeleton, exhibit movement following human anthropometry and biomechanics, and their position and posture mirror that of humans. VHs can be used in any software in a virtual environment and must have the ability to interact with and manipulate other virtual objects [18]. Virtual human interaction (VHI) should be natural and intuitive, as people prefer to interact with VHs in the same way as they interact with real people [19]. VHIs are a rapidly developing area of human–computer interaction and have found applications in various fields, such as education, healthcare, entertainment, and customer service, where VHs act as interactive/simulated tutors, patients, and assistants [20,21]. VHIs aim to simulate social interactions, communication, and engagement with entities that VHs may have, their behaviors, and responses. Previous research has shown that VHs can be used for psychological therapy, such as cognitive behavioral therapy, via an imagery focus technique [22]. Moreover, VHs present an effective tool for inducing and assessing human psychological responses [23]. De Melo [24] demonstrated that VHs can accurately assess and respond to human emotions. In addition, studies have explored the opportunities and challenges of using AI and VHs in PTA, showing that VHs can provide more standardized methods of psychological assessment [25]. Studies have examined the impact of human interaction with VHs to serve as virtual avatars on psychological and emotional responses and how these responses can be applied to psychological testing [26].

This study aims to examine the advantages and disadvantages of IVR-based role-playing with VHs for PTA through a user experience evaluation of EmpathyVR, an IVR-based empathy-type testing tool created in an interdisciplinary project. The validity of EmpathyVR for testing empathy types was demonstrated in [15]. In this study, we evaluated its user experience based on prior IVR experience, immersion, satisfaction, and embodiment in IVR role-playing scenarios involving VHs. These results evaluate the use of VHs in IVR-based role-playing for PTAs.

## 2. Background

An immersive virtual environment (IVE) is a virtual space with seamlessly integrated human–computer interaction and where immersive technology directly stimulates the user’s sensory organs through visual, auditory, tactile, and haptic modalities [27]. IVEs have provided a new direction for psychological research, particularly for understanding social and cognitive behaviors through VH interactions. IVR is increasingly being explored for applications in PTA as it provides a more interactive and controlled experimental environment than traditional self-reporting methods.

Recent studies on IVR in psychological and clinical settings highlight its potential and challenges. Kvapil Varšová and Juřík [28] noted that IVR can be used as exposure therapy to treat acrophobia, but issues like realistic user experiences and biofeedback integration remain. Research has also suggested that therapist-led sessions are more effective than automated ones [29]. Araiza et al. [30] discussed the affordability of IVR and its use in pain distraction and neuropsychological assessments, and concluded that immersion and simulation remain challenging. Riva [31] studied the long-term effects of IVR on anxiety, pain management, and eating disorders. Elor and Kurniawan [32] explored IVR in physical rehabilitation and showed that it enhanced engagement and outcomes with biofeedback. Schöne et al. [33] demonstrated that photorealistic IVR can mimic real-life experiences by showing emotional responses similar to those of real-world experiences. Quah [34] showed that IVR enhances clinical training for health students, although design issues remained. Jespersen [35] introduced the CAVIR test, an IVR tool for assessing cognitive impairment in mood or psychosis disorders, and proved its effectiveness in real-world evaluations.

Some studies have demonstrated the advantages of social presence using VHs and IVR in IVE settings, which enables social presence to be used efficiently to understand human behavior. Studies that report on relational and social behaviors are particularly noteworthy. Felnhofer [36] investigated gender and age differences in physical and social presence in an IVE. Their results revealed that men experienced more physical presence than women, but no gender differences were identified with regard to social presence. Regarding assessing empathy in IVE, Shin [37] found that the cognitive processes through which users experience quality, presence, and flow determine how they empathize with and embody IVR stories. Furthermore, they demonstrate that users’ personal traits correlate with their perceived immersion in IVE.

Recent studies have emphasized the role of VHs and IVR technologies in enhancing PTA. Lucas [38] highlighted the use of VH interviewers to reduce stigma in mental health reporting, particularly among military service members with PTSD, and suggested that VHs improve symptom disclosure through anonymity and rapport. Rizzo [39] discussed VH agents such as SimCoach and SimSensei, which enhance client interaction in healthcare by interpreting behavioral cues such as facial expressions and body gestures, thereby improving telehealth diagnostics. Gaggioli [20] explored autonomous VHs in VR therapy, emphasizing their potential to foster effective therapeutic relationships and improve patient interactions.

Previous studies investigated the use and effectiveness of role-playing in IVR in the context of psychology and training. Suárez [40] compared the effectiveness of role-playing in leadership training using VHs and real humans using three training methods: IVR, mixed reality, and conventional methods. The results revealed no significant differences between the conditions based on VHs and real humans, thus indicating the usefulness of role-playing with VHs in skill training. Sapkaroski [41] investigated the effectiveness of role-play-based communication training in a hospital setting using IVR-based role-playing with VHs and traditional role-playing conditions. Their results showed a more significant improvement in communication skills among participants using the IVR-based method compared with those using conventional methods. While these studies demonstrate the effectiveness of IVR-based role-playing with VHs for skill training, research on the user experience of IVR-based PTA solutions based on role-playing with VHs is lacking.

We assembled an interdisciplinary project team comprising psychologists, digital media creators, and computer scientists, who are domain experts and researchers. In this project, a multidimensional empathy scale for adolescents was designed by psychologists based on different empathy theories and instruments such as the Positive and Negative Empathy Scale (PaNES) [42] and the Cognitive, Affective, and Somatic Empathy Scale (CASES) [43]. Six elements along three dimensions of empathy (cognitive−affective, positive−negative, and majority−minority) were used to form eight unique types of empathy. The Multidimensional Empathy Scale for Adolescents (MESA) [44] was adapted for EmpathyVR, a scenario-based IVR role-play content for PTA, followed by a validation study [15]. However, user experiences focusing on VHI and IVR role-playing, which are important aspects of scenario-based PTA, have not been investigated. Therefore, this study examines the utility of IVR and role-playing with VHs in PTA through the user experience in EmpathyVR.

## 3. Research Question and Methodology

### 3.1. Research Questions

The mixed-method study aims to answer the following research questions:RQ 1: What aspects of IVR role-play and VHs for PTA provide positive and negative experiences for the participants?RQ 2: Do the participants’ prior IVR experience affect their sense of embodiment during role-play with VHs in EmpathyVR?RQ 3. Does the participants’ immersion level affect their satisfaction with the IVR experience during role-play with VHs in EmpathyVR?

### 3.2. Stimulus: EmpathyVR

EmpathyVR is a role-playing IVR experience containing VHs embedded in scenarios that are divided into three parts based on the number of VH friends: one friend, two friends, and several friends. The first and third parts are divided into two episodes containing opposite situations. The second part contains two friends in opposite situations. This trend is repeated for five episodes. This approach differs from the original PTA scale because the participants can be informed about the other scenarios by friends, who are played by the VHs in the IVR. The setting is a virtual high school environment with high-school characters in which the user role-plays as one of the student characters in the scenario, expressing and diagnosing the empathy type. The content starts with a title, warning, tutorial, and introduction using a virtual character. Moreover, the user’s virtual avatar is introduced through a virtual mirror so that they can see themselves and check their motor synchrony [45] by waving their hands and making facial expressions. The scenarios in the IVR role-play replace questions from the MESA scale and have been developed by psychological researchers, writers, and content designers. The PTA questions examine what the user feels or thinks about what the VH would feel in the given situation (see Appendix A). After experiencing the episodes, participants rated the questions described earlier in this chapter and explained their experiences. Table 1 summarizes the episodes and Table 2 describes the design details of the virtual environment, VHs, and interaction design in EmpathyVR.

The overall content workflow is illustrated in Figure 1. The number of episodes matched the number of episodes listed in Table 1. Q1–Q5 refer to the in stimuli questionnaires used to determine the empathy type of the users.

### 3.3. Experimental Design

To answer the research questions, we designed a mixed-method experiment comprising pre- and post-test questionnaires to collect data on the participants’ prior experience of IVR and their perceptions of the EmpathyVR tool for PTA in terms of embodiment, immersion, and overall satisfaction. The following sections describe the applied data collection instruments, participants, procedures, data analysis methods, and ethical considerations.

#### 3.3.1. Instruments

In the experiment, participants responded to a series of questionnaires that assessed various aspects of the user experience, including prior IVR experience, embodiment, immersion, and satisfaction, as well as open-ended feedback.

The quantitative statements in the pre- and post-test questionnaires were divided into the following sections.

Prior experience with IVR (pre-test): The participants were asked to evaluate their prior IVR experience on a four-point Likert scale, with the following responses: never (0), rarely (1), occasionally (2), frequently (3). This was performed to ascertain the extent of the participants’ experience with IVR before the experiment.Embodiment (post-test): This section included three questions rated on a seven-point Likert scale to measure participants’ sense of embodiment within a virtual environment [49]. The questions focused on the extent to which the participants felt that their virtual body was their own, their ability to control it as if it were real, and their perception of their body’s existence in the virtual space.Immersion (post-test): Three questions assessed the level of immersion on a seven-point Likert scale [50]. These questions evaluated the immersion of the visual aspects of the virtual environment, whether the EmpathyVR experience matched real-life experiences, and how easily the participants adapted to the virtual environment.Satisfaction with Empathy Diagnosis Content (post-test): A 10-point question asked participants how likely they were to recommend the IVR content to other middle and high-school students [51] to measure their overall satisfaction with the content. Open-ended questions were used to explore the elements of user satisfaction.

In addition to the quantitative measures, open-ended questions were included to gather qualitative data on the participants’ experiences. One positive question asked what they liked about EmpathyVR, and four negative questions explored what they found unfavorable, difficult, or uncomfortable; if they noticed any issues; and any changes they would suggest for EmpathyVR.

#### 3.3.2. Participants

The target users of EmpathyVR were Korean adolescents in secondary education. Both the MESA and IVR role-playing scenarios featured virtual school friends as VHs. The experiment was conducted at a high school in Suwon, Republic of Korea, where 99 adolescents were recruited as test participants. The participants were 15–16 years old, attending their first year of high school, and consisted of 47 females, 49 males, and 2 who identified as other.

#### 3.3.3. Procedure

The participants completed the experiment individually. A researcher greeted the participants upon arrival in the test room and explained the purpose of the experiment. The participants then completed the pretest questionnaire, following which the researcher aided them in putting on the HTC VIVE Pro Eye head-mounted display (HMD). The participant played through the EmpathyVR content and interacted with the VHs. The participants’ empathy type was determined by completing a questionnaire after each episode of the scenario. The empathy-type test result was shown to the participant at the end of EmpathyVR. The researcher assisted the participants in removing the HMD and asked them to complete the post-test questionnaire. The entire experiment lasted approximately 35 min.

#### 3.3.4. Data Analysis

We analyzed the quantitative questionnaire data using descriptive statistics (e.g., frequency, mean, standard deviation (variability), median, mode, and range). Furthermore, the chi-square test of independence was used to determine whether the key variables were dependent. The first chi-square test was conducted in the time frame of the questionnaire following the prior IVR experience (four-point frequency) and before the questionnaire on the sense of embodiment (seven-point Likert scale). The second chi-square test was conducted in the time frame between the levels of immersion (seven-point Likert scale) and satisfaction (10-point numeric scale) questionnaire. As chi-square tests of independence were conducted on categorical data, we first converted the numerical embodiment, immersion, and satisfaction data into categorical data by assigning each participant’s results to low or high groups. The division threshold for each scale was the mean score calculated from the participants’ responses. Chi-square test results were then obtained by calculating the observed and expected frequencies for each combination of categories, calculating the chi-square test statistic based on the frequencies, and checking the chi-square test result against the chi-square distribution table to determine whether the tested variables were independent.

The factors contributing to the corresponding advantages were further examined by analyzing participants’ responses to open-ended questions. Additionally, we investigated whether any aspects were identified as disadvantages or potential limitations of IVR, VHs, or role-playing. To analyze the qualitative data from the open-ended questions, we assigned a score to each key term based on the participants’ responses. First, keywords were identified, and their connotations were classified as positive or negative. Subsequently, the frequency of the appearance of each keyword within each element was calculated to generate a score.

#### 3.3.5. Ethical Considerations

The experimental plan was reviewed by the Institutional Review Board of Ajou University (202207-HS-001) before data collection. Participants and their guardians provided informed consent prior to the experiment. The collected data were handled anonymously.

## 4. Result

### 4.1. Prior Experience with IVR

The pretest questionnaire included questions about the participants’ gender, age, and prior experience with IVR. The results indicated that most participants (68%) never or rarely used IVR before the experiment, while those who occasionally or frequently used IVR represented 32% of the participants (Figure 2).

### 4.2. Embodiment and Immersion

Figure 3 shows the responses to the embodiment statement. The scores for embodiment were moderate, with most responses ranging from 4 to 5. The statement “I felt as if the virtual body was my body” (Q1) received the lowest mean (µ = 4.09, σ = 1.45) among the three statements. The statement “I felt like I could control the virtual body as if it were my own body” (Q2) had the highest mean (µ = 4.75, σ = 1.37), indicating that the participants felt a relatively strong sense of control over their virtual body. The results also suggest a wider variability in how much the participants felt that the location of their physical body corresponded to the observed location of the virtual body (Q3) (µ = 4.11, σ = 1.64). Moreover, the results for Q4 showed a mixed experience among the users, as many participants provided higher scores (5 or 6), while some provided lower ratings.

Regarding the immersion statements (Figure 4), the participants rated the ease of adaptation to the virtual environment (Q6) the highest, with a mean of 5.76 and the lowest variability (σ = 1.11). This result suggests that the participants generally found it easy to adjust to the virtual environment. The visual aspects of the virtual environment (Q4) were also rated highly (µ = 4.71, σ = 1.24), although slightly lower than the ease of adaptation (Q6). The immersion experience appeared to be consistent with the real-world experience (Q5), but this statement received a slightly lower mean (µ = 4.59, σ = 1.43) than the other statements.

The participants reported a sense of embodiment, although their connection to the virtual body was moderate. In Q2, the participants demonstrated the highest sense of control over the virtual body. Participants rated their immersive experiences highly, particularly their ability to adjust to the virtual environment, as evidenced by their responses to Q6. The visual aspects also significantly contributed to their sense of immersion. The responses to statements about embodiment exhibited more variability than the statements about immersion, particularly to Q3, which suggests that the experience of embodiment may be a more subjective or context-dependent phenomenon for the participants.

### 4.3. Satisfaction with EmpathyVR

The question “How likely do you recommend this VR content to middle and high school students?” measured the overall satisfaction of participants with EmpathyVR in its use as IVR role-playing with VHI. The results presented in Table 3 indicate a mean score of 8.31 out of 10 (σ = 1.24), thus the majority of the participants found EmpathyVR to be moderately to highly recommendable. This result suggests that EmpathyVR aligns well with the needs and expectations of the target audience.

We analyzed the participants’ responses to open-ended questions to gain further insight into their satisfaction and dissatisfaction with EmpathyVR, specifically focusing on VHI through IVR role-playing. As the qualitative data do not include specific numerical satisfaction scores but rather the frequencies of the participants who expressed likes or dislikes in various categories, we focus on analyzing the distribution and implications of these feedback categories (see Appendix B).

Participants’ satisfaction levels were determined based on the categories identified from positive feedback, whereas their dissatisfaction levels were extracted from negative feedback. The feedback covered various aspects including immersion, realism of virtual humans and their environment, usability of hardware and software, interest based on enjoyment and educational value, and opportunities for quality content to provide empathy-type detection.

Relevant comments were categorized based on their meaning and appropriate keywords were extracted. The final categories and their explanations are as follows.

Immersive: Phrases indicating a feeling of presence.Realistic Virtual Humans and Environment: Phrases describing an accurate representation similar to the real world.Usability—Hardware: Phrases signifying that the devices were useful.Usability—Software: Phrases indicating that the content fulfilled its intended purpose.Interesting and Enjoyable: Phrases suggesting that the content was entertaining.Interesting and Educational: Phrases referring to engaging and valuable content for empathy detection and learning.Opportunity Providing: Phrases indicating that the content facilitated knowledge growth and served as a platform for development and empathy detection.

To ascertain the prevalence of positive and negative experiences, one point was allocated for each positive and negative connotation of the keyword, respectively. This approach enabled the systematic quantification of qualitative feedback from the open-ended questions.

Figure 5 illustrates the frequencies of the aspects in EmpathyVR on which the participants expressed positive and negative feedback on the identified categories. Figure 6 provides additional detailed information on the positive and negative responses related to VHs and the environment.

Notably, participants highlighted the realistic environment in positive responses, while they expressed reservations about the realism of VHs in negative responses.

### 4.4. Relationship Between Familiarity with IVR and Embodiment

For further investigation, a chi-square test was conducted to examine the following hypothesis for answering Research Question 2:


*The participants’ prior IVR experience is independent of their embodiment level during role-play with VHs in EmpathyVR.*


To examine the relationship between prior experience with IVR and the level of embodiment during EmpathyVR, each element of the frequency of experience was scored on a scale of 1 to 4, with higher numbers indicating more frequent use and relative familiarity with IVR. We assigned scores of 1 and 2 to the low-experience group, and scores of 3 and 4 to the high-experience group. Similarly, we assigned the embodiment results (three statements on a seven-point Likert scale) to the low- and high-embodiment groups by comparing the sum of the statement scores to the mean of all participants’ embodiment scores. Thirty-two participants reported having occasional IVR, while 67 participants reported having rarely experienced IVR. Of the participants who occasionally-frequently used IVR, 19 reported a high level of embodiment and 13 reported a low level of embodiment. Conversely, of the participants who rarely or never used IVR, 44 reported a high level of embodiment and 23 reported a low level of embodiment. Table 4 lists these results along with the results of the chi-square test. Because the *p*-value was 0.700, we failed to reject the null hypothesis. This result indicates that there is no statistically significant relationship between the participants’ prior IVR experience and their embodiment levels; thus, these variables are independent.

### 4.5. Relationship Between Immersion and Satisfaction

Another chi-square test was conducted with the following hypothesis to answer Research Question 3:


*The participants’ immersion and satisfaction levels are independent of each other during role-play with VHs in EmpathyVR.*


To examine the relationship between immersion and satisfaction, we first converted the numerical data on immersion and satisfaction into categorical data by comparing each participant’s score to the mean of all the participants’ scores. Scores below or above the mean were assigned to the low and high groups, respectively. We then performed a chi-square test on the two variables to determine whether the immersion and satisfaction levels were independent (Table 5). The chi-square test of independence results in Table 5 suggest that we can reject the null hypothesis because the *p*-value (0.0036) is less than 0.05, which is indicative of a statistically significant relationship between immersion and satisfaction.

## 5. Discussion

Previous research has demonstrated the effectiveness of role-playing with VHs for skill training; however, a significant gap remains in understanding the user experience of IVR-based PTA solutions that incorporate role-playing with VHs [41,52]. To address this gap, an interdisciplinary team of psychologists, digital media creators, and computer scientists collectively examined the effectiveness of IVR-based role-playing with VHs in PTA through an analysis of user experience in a scenario-based EmpathyVR IVR experience. Quantitative data were analyzed using descriptive statistics and chi-square tests to examine the relationships among IVR immersion, VH experience, prior VR experience, embodiment, and content satisfaction. Open-ended responses were evaluated by identifying key terms, classifying them as positive or negative, and calculating their frequencies to assess the advantages, limitations, and user perceptions of IVR, VHs, and role-playing in PTA. The following sections present our interpretations of the results along with the implications and limitations of the study.

### 5.1. Advantages and Disadvantages of Role-Playing with VHs in IVR-Based PTA

Participants demonstrated a mixed sense of embodiment. While users reported a relatively strong ability to control the virtual body (Q2) with a mean score of 4.75, their overall connection to the virtual body was moderate, as evidenced by the lower scores in other embodiment-related statements. The variability in the embodiment scores, especially in Q3, with a mean of 4.11 and a standard deviation of 1.65, suggests that the feeling of embodiment is subjective. Some participants felt more embodied than others, perhaps owing to differences in their prior IVR experiences, personal preferences, or engagement with the scenario.

In contrast, the participants reported higher levels of immersion more consistently, particularly in terms of how quickly they adapted to the IVR environment. The mean score (5.76) and low variability (σ = 1.11) for the adjustment statement highlight that the participants generally found it easy to immerse themselves in the IVE. This response aligns with the high ratings for the visual aspects of the environment (Q4) (µ = 4.71, σ = 1.25). Notably, the participants rated their immersion as consistent with real-world experiences (Q5), although this rating was slightly lower (µ = 4.59, σ = 1.44) than the adaptation and their visual aspects of the virtual environment. This trend indicates that, while the virtual environment was immersive, some participants may still have felt certain limitations in how closely it mimicked real-world scenarios.

Furthermore, Figure 5 and Figure 6 illustrate that the lack of realism of VHs can present a negative experience in IVR role-playing. This can be attributed to quality issues in the VH animations and the lack of realism in the real-time rendering of the VHs caused by the limitations of IVR hardware, where the uncanny valley phenomenon is evident. However, this did not significantly impact the validity of PTA or satisfaction with the content (Table 3 and Figure 5).

In conclusion, role-playing with VHs could provide advantages for PTA, and the nature of immersive situations and interesting stories might introduce great potential for future PTA. Empathy can appeal to a user’s social desirability and allow the user to find its usefulness. However, the embodiment of the user’s virtual body may be subjective, depending on the user’s perspective, thus suggesting the need for a personalized embodiment experience.

Furthermore, this study showed that the lack of realism in VHs and the quality of the VHI could diminish users’ interest, as indicated by the high scores shown in Figure 5 and Figure 6. Despite the negative evaluation of the VHs and hardware-related complications (Figure 5), no significant disadvantages were reported regarding overall satisfaction.

### 5.2. Prior IVR Experience and Embodiment

The chi-square test results between prior IVR experience and embodiment did not reveal a significant dependency between the variables. This finding suggests that no statistically significant correlation exists between prior experience with IVR and the embodiment in a dataset. This result suggests that familiarity with IVR technology does not affect the users’ sense of connection with their virtual bodies. It can be reasonably assumed that users’ prior experience aids them in navigating and interacting within a virtual environment. Nevertheless, the findings indicate that even participants with limited experience demonstrated moderate to high levels of embodiment, especially regarding control over the virtual body, thereby suggesting that the design and content of EmpathyVR effectively evoked a sense of embodiment, even among those with minimal experience with IVR.

### 5.3. Level of Immersion and Satisfaction

The high mean satisfaction score of 8.31 out of 10 and the results of the qualitative data analysis indicated that most participants found IVR-based role-playing with VHs to be immersive, interesting, enjoyable, and educational. Furthermore, the chi-square test results for the levels of immersion and satisfaction revealed a significant dependency between the variables. Participants who exhibited higher immersion scores consistently reported higher satisfaction with the IVR role-playing content, whereas those exhibiting lower immersion scores correlated with lower satisfaction. This trend indicates that the immersive quality of role-playing with VHs in the IVR environment significantly shaped the overall satisfaction of the user.

### 5.4. Implications

This study demonstrated that IVR-based role-playing with VHs can provide a positive user experience in PTA. Using scenarios that reflect real-world social situations through the VHI likely contributed to the high levels of immersion and satisfaction reported by the participants. This positive response suggests that combining IVR, role-playing, and VHs could offer a new frontier in PTA by providing more dynamic and interactive methods for assessing psychological states. We expect that the quality and realism of VHs can be enhanced alongside the development of new IVR technology to enable higher resolutions and frame rates, which are essential for producing realistic IVEs rendered in real time. Furthermore, recent advances in artificial intelligence, especially in terms of large language models capable of reasoning and mimicking emotional intelligence, have enabled the development of VHs capable of personalizing interactive PTA experiences to match user expectations, preferences, and needs.

In the user study, we used a questionnaire to measure the sense of embodiment. However, Guy et al. [53] pointed out in their literature review on embodiment and its assessment methods that questionnaires may not be optimal for measuring embodiment. Instead, researchers should utilize physiological sensors, particularly electroencephalography, which enables real-time measurement of embodiment in virtual environments [53]. Moreover, electroencephalography can be used to implement brain–computer interfaces [54], thus opening avenues for innovation in human–computer interactions and VHI.

### 5.5. Limitations

This study is a derivative of the effectiveness study of EmphathyVR and was not designed to fully test the feasibility of IVR and role-playing with VHs in PTAs. Accordingly, a somewhat redundant evaluation methodology was employed to assess the user experience of the content. The data were processed by converting the scores to the contextual positivity of the keywords that correspond to IVR and role-playing with VHs in the subjective satisfaction questionnaire. Multiple researchers crosschecked the scores to reduce the likelihood of errors during score conversion. However, it must be acknowledged that this method is not optimal. Additionally, given that EmpathyVR is designed to assess empathy, it is plausible that role-playing with VHs was more effectively utilized and rated higher in terms of satisfaction because it prompts users to consider the feelings of others. To address this potential bias, it would be beneficial to expose users to situations, such as the Myers–Briggs-Type Indicator (MBTI), to explore users’ perceptions and attitudes toward PTAs in a similar manner.

## 6. Conclusions

The results of this study shed light on the potential of IVR and VHI in PTA. The participants showed high levels of immersion, particularly in adapting to the virtual environment, whereas their experiences with embodiment exhibited more variation. Satisfaction was strongly linked to immersion, and participants with prior IVR experience overall reported higher levels of embodiment. These findings suggest a significant application of IVR-based PTA, including role-playing with VHs. However, further work is required to optimize the sense of embodiment to fully realize the potential of these immersive environments.

Future work should address the suboptimal appearance and animations of VHs in EmpathyVR (Figure 5 and Figure 6), which were rated highly, but may not be sufficient to overcome the uncanny valley effect [55]. A realistic alternative is to balance quality and optimization, such as making VHs completely realistic or using more stylized characters. Although the shortcomings of VHs pose a potential threat to immersion, this study and previous research have shown that these issues do not significantly affect the immersion levels, usability, or effectiveness of PTA [15].

The answer to RQ2 reveals the following insights into the future of IVR-based PTA. The growth in IVR users will provide a greater scope for observing changes in user psychology and behavior in an immersive environment conducive to PTA. In addition, it would be beneficial to determine whether the accuracy of diagnosis can be improved and predicted over time through objective measurement techniques, such as physiological sensors, and whether in-depth analysis beyond diagnosis by scale is possible. An important concern regarding the implementation of a PTA through IVR-based role-playing with VHs is the monitoring of user behavior using sensors.

## Figures and Tables

**Figure 1 sensors-25-02719-f001:**
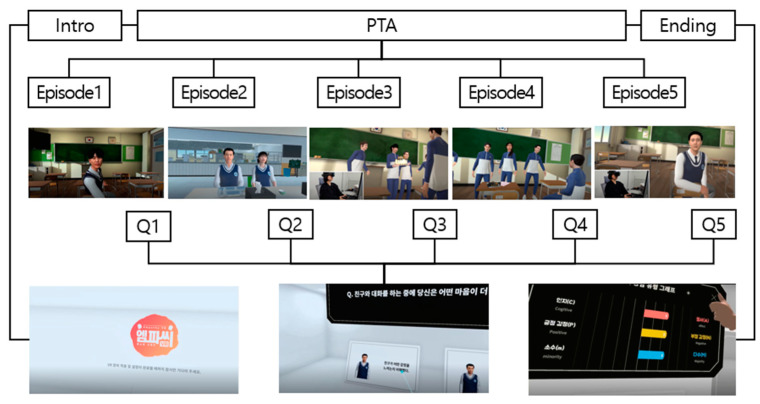
EmpathyVR content flow (bottom left: splash screen showing the EmpathyVR logo in Korean; bottom center: a partial question in Korean about what the character feels when they talk to the friend; bottom right: empathy type test results indicating that the user has tendency toward affective, negative, and majority empathy types).

**Figure 2 sensors-25-02719-f002:**
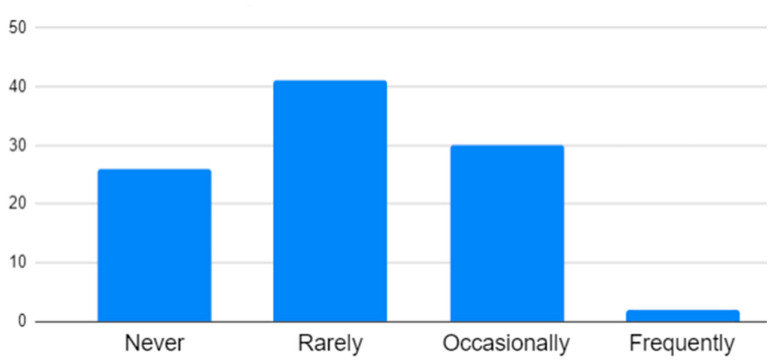
Pre-survey: Q3. How often have you used VR before this evaluation?

**Figure 3 sensors-25-02719-f003:**
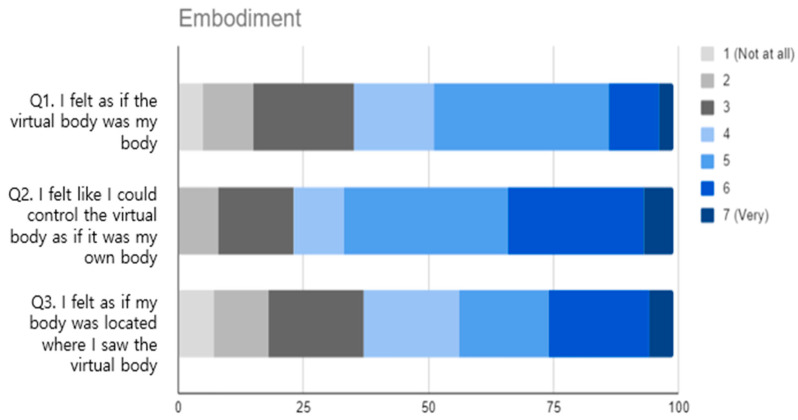
Results of the embodiment statements in the post-test questionnaire.

**Figure 4 sensors-25-02719-f004:**
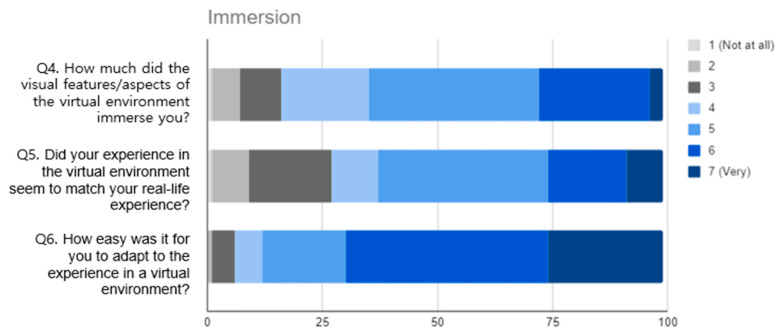
Results of the immersion statements in the post-test questionnaire.

**Figure 5 sensors-25-02719-f005:**
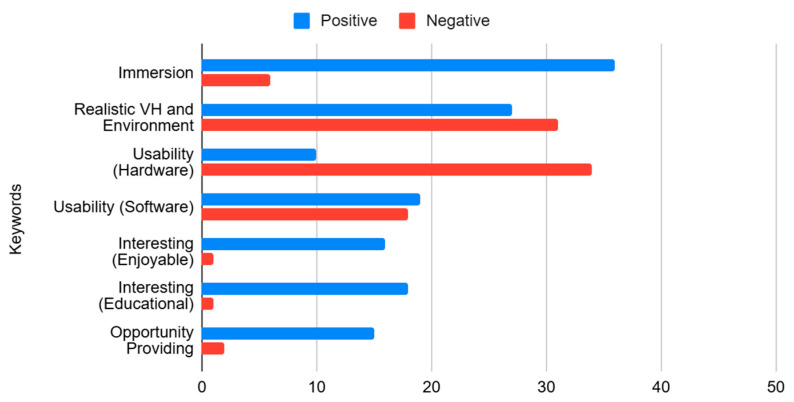
Categorized scoring of participants’ responses to the post-test questions: “What did you like about the VR content?” and “What was bad about the VR content?”.

**Figure 6 sensors-25-02719-f006:**
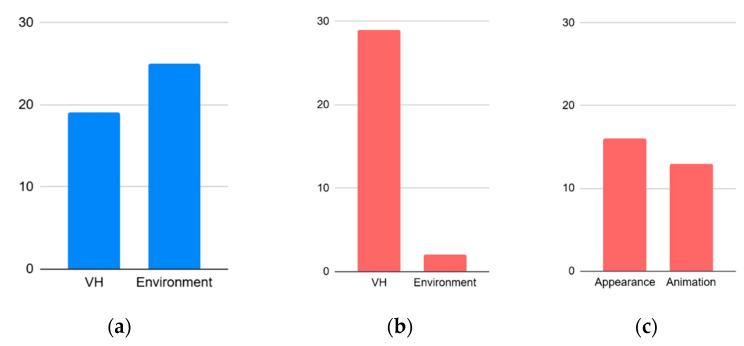
Results of responses: (**a**) Positive responses of participants for VH and environment post-test questions: “What did you like about the VR content?”; (**b**) Negative responses of participants for VH and environment “What was bad about the VR content?”; (**c**) Negative responses of participants on VH regarding appearance and animation.

**Table 1 sensors-25-02719-t001:** IVR scenarios and episodes.

VHs	Process	Story	Scene Image
Avatar	Virtual self [46] check in a mirror (male and female)	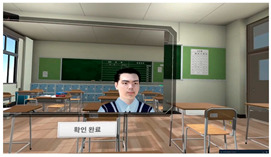 (button text: complete verification) 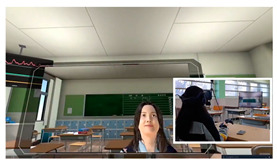	
One VH	Episode1. Positive	A boy has a crush on a girl and she likes him too.	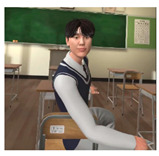
	Episode5. Negative	A boy was rejected by one of the girls he liked.	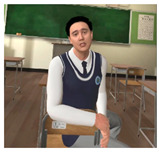
Two VHs	Episode2. Positive and negative	A boy who lost and a girl who won in the badminton tournament.	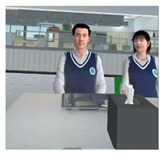
Several VHs	Episode4. Positive for the majority vs. negative for the minority	A grumpy student who lost his phone, and a happy group for celebrating someone’s birthday.	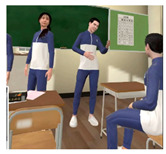
	Episode3. Positive for the minority vs. negative for the majority	A group of students lost a soccer game; a boy won a medal in a running competition.	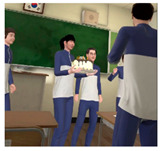

**Table 2 sensors-25-02719-t002:** EmpathyVR world and interaction details.

Category	Details
World Design	
Environment	Classroom, school cafeteria
Virtual Humans (VH)	11 virtual humans (VHs) performing 14 roles in the scenario
VH Creation	3D digital humans created through image-based modeling in Character Creator v4.5 (Reallusion Inc., San Jose, CA, USA), using photographs of staff members in their early 20s
Animation	Character animations were recorded with actors using the Perception Neuron 3 motion capture system and AxisStudio 2.8 (Noitom Ltd., Beijing, China). The captured motion data was refined and processed in Maya 2021 (Autodesk Inc., San Francisco, CA, USA) and then applied to the characters in iClone 8 (Reallusion Inc., San Jose, CA, USA), with additional clean-up, polishing, and post-animation work. Missing animations were added, and unnecessary parts were removed to enhance realism and immersion.
**Tools**	
Character Creator 4.5 [47]	Character creation tool from Reallusion Inc. Version: 4.5 The software generates and customizes realistic 3D characters that are used across multiple 3D modeling platforms and game development engines. Scientific rationale: We tested Unreal Engine’s MetaHuman with FaceAR, Photogrammetry, Face Builder for Blender, and Headshot in Character Creation to generate an image-based 3D face model. Headshot in Character Creation was selected owing to its ability to quickly create realistic characters and its compatibility across multiple platforms. It is suitable for efficient integration into diverse 3D environments, as well as being fast, affordable, and less dependent on the skill of the user.
Perception Neuron 3 [48]	Motion capture system from Noitom Ltd. (Beijing, China). Version: 3.0 A full-body motion tracking system with 23 wireless inertial measurement sensors that are mounted on a custom suit, which the actor must equip while recording motions. Scientific rationale: We compared the HTC VIVE Tracker 3.0, Rococo Vision, and Perception Neuron 3 as potential motion capture technologies. Perception Neuron 3 tool was selected because of its portability, compact design, and cost-effectiveness compared to the other motion capture technologies. Moreover, it allows seamless transfer of recorded motion data to digital character animations.
**Interaction Design**	
Object manipulation	The user interacts with the world and objects using the IVR controllers, which enable basic actions such as grabbing, rotating, pointing, and clicking.
Seated IVR and control	Although EmpathyVR is used in a seated position, the user can move their upper body, hands, and shoulders, and rotate their head in IVR based on their physical movements.
Device and specifications	HTC VIVE Pro Eye (HTC Corporation, Taoyuan, Taiwan) was used for the HMD IVR display and to track participants’ head and eye movements. HTC VIVE Trackers (HTC Corporation, Taoyuan, Taiwan) were used to track the participants’ upper body and arm movements.
Empathy-type evaluation	After exploring each episode, the user answers questions (based on MESA), and their empathy type result is computed based on their responses.

**Table 3 sensors-25-02719-t003:** Results of the user satisfaction statement (“How much do you recommend this VR content to middle school and high school students?”).

	Mean	Median	Mode	Standard Deviation	Min	Max	Range	25th Percentile	50th Percentile	75th Percentile
Value	8.31	8.00	8.00	1.24	4.00	10.00	7.00	7.50	8.00	9.00

**Table 4 sensors-25-02719-t004:** Results of the chi-square test of independence between familiarity with IVR level and satisfaction level (significance level = 0.05).

Experience/Embodiment		High Embodiment		Low Embodiment	Total
**High Experience**		19		13	32
**Low Experience**		44		23	67
**Total**		63		36	99
**Chi Square Test** **Result**	Chi Square Statistic	*p*-Value	Degrees of freedom	Expected frequencies	
	0.149	0.700	1.0	[[20.36, 11.64], [42.64, 24.36]]	

**Table 5 sensors-25-02719-t005:** Results of the chi-square test of independence between the immersion level and satisfaction level (significance level = 0.05).

Immersion/Satisfaction		High Satisfaction		Low Satisfaction	Total
**High Immersion**		50		6	56
**Low Immersion**		24		19	43
**Total**		74		25	99
**Chi Square Test ** **Result**	Chi Square Statistic	*p*-Value	Degrees of freedom	Expected frequencies	
	12.72	0.000362	1.0	[[41.86, 14.14], [32.14, 10.86]]	

## Data Availability

The data will be available from the corresponding author upon a reasonable request.

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
