# Peer review of "User Experience of Virtual Human and Immersive Virtual Reality Role-Playing in Psychological Testing and Assessment: A Case Study of ‘EmpathyVR’"

_sensors, 2025, doi:10.3390/s25092719_

Round 1

Reviewer 1 Report

Comments and Suggestions for Authors

The paper presents interesting and timely research investigating the user experience of role-playing with VHs in the context of IVR-based psychological testing and assessment (PTA), focusing on EmpathyVR, an IVR-based empathy type assessment tool previously developed. Specifically, the research aims to evaluate the advantages and disadvantages of integrating IVR-based role-playing with VHs into PTA by examining users’ immersion, embodiment, and satisfaction.

Notwithstanding the importance of the research and the results, which, though not conclusive, offer a good starting point for further analysis in the field, I have some comments, which are depicted below.

  1. The paper's goals should be better described because the description is often too vague.
  2. This also affects the 2 Research Questions. Specifically, RQ2 is too ambiguous, and it is thus hard to understand if the results support it or are in contrast with it. I suggest splitting RQ2 into different sub-questions and analyzing data accordingly.
  3. Related to 2, but also valid in general, the methods that support the results in Subsection 4.5 are not sufficiently explained. The authors mention the chi-square methods, but more details on how data are analyzed is necessary, to understand Table 5 and related discussions.
  4. Embodiment is one of the main discussion points because the results are somehow mixed up. For this reason, the authors should clarify the experimental methodology better. Did the subject have a period of time to elicit embodiment? Are the subjects able to see their own body somehow (e.g., with a mirror)? Did the subjects have the chance to choose their representation? How was tracking and animation of the body achieved? 

Author Response

1. The paper's goals should be better described because the description is often too vague.

Firstly, thank you for your valuable and thoughtful feedback that helped us significantly improve the manuscript!

As mentioned in the Abstract, this research aimed to evaluate the advantages and disadvantages of integrating IVR-based role-playing with VHs into PTA by examining users’ immersion, embodiment, and satisfaction. However, the goal of this research was not described well in the context of the introduction. Thus, we added text at the end of the Introduction.

2. This also affects the 2 Research Questions. Specifically, RQ2 is too ambiguous, and it is thus hard to understand if the results support it or are in contrast with it. I suggest splitting RQ2 into different sub-questions and analyzing data accordingly.

We agree that the previous RQ2 covered two different aspects, so we split it into two questions:

    • RQ 2: Do the participants’ prior IVR experience affect their sense of embodiment during role-play with VHs in EmpathyVR?
    • RQ 3. Does the participants’ immersion level affect their satisfaction with the IVR experience during role-play with VHs in EmpathyVR?

3. Related to 2, but also valid in general, the methods that support the results in Subsection 4.5 are not sufficiently explained. The authors mention the chi-square methods, but more details on how data are analyzed is necessary, to understand Table 5 and related discussions.

We appreciate your appropriate questions regarding the research question and the methodology used to find the answers. By splitting the previous RQ 2 into two sub-questions (see previous answer), the hypotheses according to those two questions in sections 4.4 and 4.5 now make more sense. Moreover, we added further explanation of the analysis process, particularly related to the analysis of the chi-square tests of independence, to Section 3.3.4. Sections 4.4. and 4.5 presenting the results of the chi-square tests were also amended to clarify the results.

    • Page 9 “Furthermore, the chi-square test of independence was used to determine whether the key variables were dependent. The first chi-square test was conducted in the time frame of the questionnaire following the prior IVR experience (four-point frequency) and before the questionnaire on the sense of embodiment (seven-point Likert scale). The second chi-square test was conducted in the time frame between the levels of immersion (seven-point Likert scale) and satisfaction (10-point numeric scale) questionnaire. As chi-square tests of independence were conducted on categorical data, we first converted the numerical embodiment, immersion, and satisfaction data into categorical data by assigning each participant’s results to low or high groups. The division threshold for each scale was the mean score calculated from the participants’ responses. Chi-square test results were then obtained by calculating the observed and expected frequencies for each combination of categories, calculating the chi-square test statistic based on the frequencies, and checking the chi-square test result against the chi-square distribution table to determine whether the tested variables were independent.”
    • Page 13 and 14: We have described the analysis in detail.

4. Embodiment is one of the main discussion points because the results are somehow mixed up. For this reason, the authors should clarify the experimental methodology better. Did the subject have a period of time to elicit embodiment? Are the subjects able to see their own body somehow (e.g., with a mirror)? Did the subjects have the chance to choose their representation? How was tracking and animation of the body achieved? 

Yes, there was a part in the IVR experience to allow the participants observe their virtual avatar in a virtual mirror. We explained the scene where participants check their motor synchrony with the avatars using a phrase that cites a reference. We added the following scene images to Table 1 and an explanation of the procedure in Section 3.2. Stimulus: EmpathyVR.

    • Page 5: “The content starts with a title, warning, tutorial, and introduction by a virtual character. Moreover, the user’s virtual avatar is introduced through a virtual mirror so that they can see themselves and check their motor synchrony [46] by waving their hands and making facial expressions.”

We hope our modification satisfies you.

Reviewer 2 Report

Comments and Suggestions for Authors

While the manuscript explores an engaging interdisciplinary topic, the authors should ensure the study clearly aligns with the journal's scope, particularly by elaborating on the sensing technologies involved. All acronyms (e.g., IVR, VH, PTA) should be defined at first use, and an acronym list is recommended. The description of tools such as Character Creator (Reallusion) and Perception Neuron 3 should include versions, core features, and scientific rationale. The methodology would benefit from more clarity on measurement instruments and statistical analysis, especially concerning embodiment variability. Some references cited in-text are missing from the reference list and should be updated per journal style, with more recent literature included (past five years). The manuscript could be streamlined by combining or condensing some tables and appendices, and the article type might be better categorized as a "Case Report" per MDPI guidelines. I hope these suggestions help support a clearer and more scientifically robust revision.

Comments on the Quality of English Language

The English could be improved to more clearly express the research.

Author Response

While the manuscript explores an engaging interdisciplinary topic, the authors should ensure the study clearly aligns with the journal's scope, particularly by elaborating on the sensing technologies involved.

Thank you for your insightful and valuable comments, which helped us significantly improve the manuscript! We understand your concern. However, this special issue, ‘Virtual Reality and Sensing Techniques for Human, ’ covers a broad range of HCI, as shown in the topics of interest. Our manuscript meets ‘VR-based interaction design,’ ‘Embodiment and presence,’ and ‘Social interaction in VR.’

Virtual reality (VR) has evolved as a transformative platform for developing immersive and interactive experiences, enabling new ways for people to interact with digital content, environments, and one another. Expertise in computer graphics, human–computer interaction, cognitive psychology, sensor technology, and other fields is combined in this multidisciplinary field. With the goal of improving the quality, realism, and efficacy of human interaction in virtual environments, this Special Issue is looking for ground-breaking research, cutting-edge methodologies, and real-world applications that explore the relationship between VR and sensing techniques. Topics of interest include, but are not limited to:

  • VR-based interaction design;
  • Multisensory experiences;
  • Sensor fusion for VR;
  • Embodiment and presence;
  • Social interaction in VR;
  • Ethical and privacy considerations;
Health and well-being applications.

All acronyms (e.g., IVR, VH, PTA) should be defined at first use, and an acronym list is recommended.

We also added the abbreviations at the end of the document just before the appendices.

    • Page 17

"Abbreviations

IVR

IVE

VH

PTA

VHI

PaNES

CASES

MESA

HMD

MBTI

MSIT

ITRC

IITP

Immersive virtual reality

Immersive virtual environment

Virtual Human

Psychological testing and assessment

Virtual Human Interaction

Positive and Negative Empathy Scale
Cognitive, Affective, and Somatic Empathy Scale

Multidimensional Empathy Scale for Adolescents

Head-mounted display

Myers-Briggs type indicator

Ministry of Science and ICT

Information Technology Research Center

Institute for Information & Communications Technology Planning & Evaluation

The following abbreviations are used in this study:"

The description of tools such as Character Creator (Reallusion) and Perception Neuron 3 should include versions, core features, and scientific rationale.

We have added the details of the tools and the reasons we chose them. We have tested several software applications and motion sensors to find the best match for VH and motion generation in the EmpathyVR project.

    • Page 6 and 7

Character Creator 4.5 [48]

Character creation tool from Reallusion Inc. 

Version: 4.5

It facilitates generating and customizing realistic 3D characters utilized across multiple 3D modeling platforms and game development engines.

Scientific rationale: We tested Unreal Engine’s MetaHuman with FaceAR, Photogrammetry, Face Builder for Blender, and Headshot in Character Creation to generate an image-based 3D face model. Headshot in Character Creation was selected due to its ability to quickly create realistic characters and its compatibility across multiple platforms. It is suitable for efficient integration into diverse 3D environments, being fast, affordable, and less dependent on the designer's skill.

Perception Neuron 3 [49]

Motion capture system from Noitom.

Version: 3.0

A full-body motion tracking system with 23 wireless inertial measurement sensors mounted on a custom suit that the actor must equip while recording motions.

Scientific rationale: We compared the HTC VIVE Tracker 3.0, Rococo Vision, and Perception Neuron 3 as potential motion capture technologies. Perception Neuron 3 tool was selected due to its portability, compact design, and cost-effectiveness compared to the other motion capture technologies. Moreover, it allows seamless transfer of recorded motion data to digital character animations.

The methodology would benefit from more clarity on measurement instruments and statistical analysis, especially concerning embodiment variability.

We fully understand your concerns. To evoke the user's embodiment, we have the process of self-checking in a virtual mirror, added the images in Table 1, and described the procedure.

    • Indeed, there was a process of seeing the participants check their virtual selves in a mirror. So, we added the scene images in Table 1 in Section 3.2 Instrument and the procedure.

    • Page5 “The content starts with a title, warning, tutorial, and introduction by a virtual character. Moreover, there is a process to introduce the user’s virtual avatar through a virtual mirror so that the user can see themselves and check their motor synchrony [46] by waving their hands and making facial expressions.”
    • Furthermore, we have amended sections 3.3.4, 4.4, and 4.5 to clarify how the chi-square tests were conducted.

Embodiment variability was measured by standard deviation. We noted this in section 3.3.4 by adding the word “variability” in conjunction with standard deviation.

Some references cited in-text are missing from the reference list and should be updated per journal style, with more recent literature included (past five years).

Indeed, a couple of references were missing. We appreciate your dedicated and sharp-eyed review. We added the missing references to our manuscript.

Regarding the references, we have included recent technical journal papers related to embodiment, but kept older papers on fundamental psychological theories in the updated version.

    • The list of added references of recent literature:

Kim C-S, Jung M, Kim S-Y and Kim K 2020 Controlling the Sense of Embodiment for Virtual Avatar Applications: Methods and Empirical Study JMIR Serious Games 8 e21879

Dewe H, Gottwald J M, Bird L-A, Brenton H, Gillies M and Cowie D 2022 My Virtual Self: The Role of Movement in Children’s Sense of Embodiment IEEE Trans. Visual. Comput. Graphics 28 4061–72

Guy M, Normand J-M, Jeunet-Kelway C and Moreau G 2023 The sense of embodiment in Virtual Reality and its assessment methods Front. Virtual Real. 4 1141683

Nwagu C, AlSlaity A and Orji R 2023 EEG-Based Brain-Computer Interactions in Immersive Virtual and Augmented Reality: A Systematic Review Proc. ACM Hum.-Comput. Interact. 7 1–33

The manuscript could be streamlined by combining or condensing some tables and appendices, and the article type might be better categorized as a "Case Report" per MDPI guidelines.

We appreciate your concern. However, after reviewing all article types listed by MDPI (https://www.mdpi.com/about/article_types), we conclude that our manuscript matches the “Article” type because it presents original research as an experiment. Case Report, as per our understanding, is more appropriate for medical studies that “present detailed information on the symptoms, signs, diagnosis, treatment (…), and outcomes of an individual patient.” 

I hope these suggestions help support a clearer and more scientifically robust revision.

According to a reviewer's recommendation, we had an English proofreading session with a qualified expert. We hope the reviewer can see the improvement in our manuscript language.

Round 2

Reviewer 2 Report

Comments and Suggestions for Authors

Accept in present form